# Retrospective Assessment of The Intestinal Protozoan Distribution in Patients Admitted to The Hospital Aristide Le Dantec in Dakar, Senegal, from 2011 to 2020

**Mouhamadou Ndiaye** [1,2,*], **Khadim Diongue** [1,2], **Mame Cheikh Seck** [1,2], **Mamadou Alpha Diallo** [2],
**Ekoué Kouevidjin** [1], **Aida Sadikh Badiane** [1,2] and **Daouda Ndiaye** [1,2]

1   Laboratory of Parasitology and Mycology, Cheikh Anta Diop University, Avenue Cheikh Anta Diop, Fann, BO 16477 Dakar, Senegal
2   Laboratory of Parasitology and Mycology, Aristide Le Dantec Teaching Hospital, BP 5005 Dakar, Senegal
*   Correspondence: mouhamadou.ndiaye@ucad.edu.sn; Tel.: +22-1775744495

**Abstract:** Infectious parasites, especially the intestinal protozoan parasites, continue to be a major public health problem in Africa, where many of the same factors contribute to the transmission of these parasites. This study was conducted to investigate the parasites causing intestinal protozoal infections diagnosed in Aristide Le Dantec hospital (Senegal). Direct examination and the Ritchie technique were used. Among the 3407 stool samples studied, 645 demonstrated the presence of intestinal protozoa in single parasitism, biparasitism, or polyparasitism, representing a prevalence of 18.93%. Out of a total of 645 protozoa, 579 (16.99%) were identified in monoparasitism in the following order: *Entamoeba coli* (6.87%) and *Blastocystis hominis* (5.69%) for low pathogenic species, and *Entamoeba histolytica/dispar* (2.31%) and *Giardia intestinalis* (1.32%) for pathogenic species. The rates of biparasitism and polyparasitism were 1.88% and 0.06%, respectively. The highest rate of parasites was 24.83% between the ages of 0–15 years. A logistical regression model indicated that intestinal protozoan infections were not associated with age groups. There was an association between age groups and *Giardia intestinalis* and *Blastocystis hominis* ($p < 0.05$). These results demonstrated the frequency of intestinal protozoa in Senegal. There is a need to implement treatment, prevention, and control measures to limit the circulation of these protozoan infections.

**Keywords:** *Entamoeba histolytica*; Giardia intestinalis; protozoan intestinal; hygiene; stool samples

## 1. Introduction

Intestinal parasites are the organisms that live in the intestine at the cost of their host. Characteristics such as social customs, religious affairs, environmental factors, availability of intermediate hosts, familiar habits, and personal hygiene of individuals affect the distribution of parasites [1]. More than 3 billion people worldwide are being infected with various intestinal parasites, leading to morbidity in 450 million individuals [2]. The prevalence of these infections is influenced by geographic, neighborhood, behavioral, biological, and socioeconomic considerations. These infections are strongly associated with the rainy tropical climate, limited access to potable water, poor environmental sanitation, overcrowding, and low family income. All these factors promote and facilitate the growth, transmission, and access to intestinal parasites [3]. Both intestinal helminths and protozoan infections have been reported as major contributors to disease and mortality worldwide [4]. The most prevalent enteric protozoa are *Entamoeba histolytica/dispar*, *Cryptosporidium* spp., *Giardia. intestinalis*, *Blastocystis hominis*, *Cyclospora cayetanensis*, and *Cystoisospora belli.* A number of agents are responsible for diarrheal diseases, among which protozoa intestinal parasites are important providers that can be transmitted through the ingestion of contaminated food and water [5,6]. Protozoal intestinal infections are described as chronic to severe

diarrhea, sometimes with abdominal cramps, flatulence, nausea, vomiting, anorexia, tiredness, low-grade fever, and weight loss [7–9]. To achieve the goal of eliminating intestinal parasitic infections as a public health problem, the WHO has suggested the mass administration of a single oral dose of mebendazole or albendazole administered periodically to preschool and school-aged children living in endemic areas. This is an intervention that reduces morbidity by decreasing the vermin burden [10,11]. There are few or no effective drug treatments for intestinal protozoa. Treatment of Giardia and amoebae is based on 5-nitroimidazole derivatives. Single-dose treatments can be administered with tinidazole or secnidazole [12].

In Senegal, intestinal protozoan infections are prevalent due to poor environmental and personal health and contamination of food and water due to unsanitary disposal of human and animal feces [10–15]. In our present study, the objective was to show the distribution of intestinal protozoa detected in patients who were referred to the parasitology laboratory of the hospital Aristide Le Dantec in Dakar between January 2011 and December 2020 and the relationship of this distribution with variables such as age, sex, season, and hospitalization or non-hospitalization status.

## 2. Results

### 2.1. Description of the Study Population

The patient demographics of all of the study participants are summarized in Table 1. A total of 3407 patients were included in the study, with a sex ratio of 1.04. The age of the patients ranged from 4 months to 91 years with a mean age of 35.6 years. The distribution of patients by age category was as follows: 0–15 years, 302 (8.86% (95% CI: 07.95–09.87)); 15–30 years, 1139 (33.43% (95% CI: 31.87–35.03)); 31–60 years, 1562 (45.85 % (95% CI: 44.18–47.52)), and > 60 years, 404 (5.52% (95% CI: 10.81–12.99)).

**Table 1.** Socio-demographic profiles of patients in a stool sample analysis from Dakar, Senegal, in the years 2011–2020.

| | **Number** | **Percentage** | **CI 95%** |
|---|---|---|---|
| Years | | | |
| 2011 | 408 | 11.98 | (10.93–13.11) |
| 2012 | 437 | 12.83 | (11.74–13.99) |
| 2013 | 344 | 10.1 | (09.13–11.15) |
| 2014 | 313 | 9.19 | (08.26–10.20) |
| 2015 | 292 | 8.57 | (07.68–09.56) |
| 2016 | 416 | 12.21 | (11.15–13.35) |
| 2017 | 414 | 12.15 | (11.10–13.29) |
| 2018 | 343 | 10.07 | (09.10–11.12) |
| 2019 | 252 | 7.4 | (06.56–08.33) |
| 2020 | 188 | 5.52 | (04.80–06.34) |
| Age group | | | |
| <15 yrs | 302 | 8.86 | (07.95–09.87) |
| 15–30 yrs | 1139 | 33.43 | (31.87–35.03) |
| 31–60 yrs | 1562 | 45.85 | (44.18–47.52) |
| >60 yrs | 404 | 11.86 | (10.81–12.99) |
| Gender | | | |
| Male | 1735 | 50.92 | (49.24–52.60) |
| Female | 1672 | 49.08 | (47.40–50.76) |
| Service | | | |
| Hospitalized | 896 | 26.3 | (24.85–27.8) |
| Non-hospitalized | 2511 | 73.7 | (72.2–75.15) |

**Table 1.** *Cont.*

|  | Number | Percentage | CI 95% |
|---|---|---|---|
| Seasons | | | |
| Dry | 2814 | 82.59 | (81.28–83.83) |
| Rainy | 593 | 17.41 | (16.17–18.72) |
| Intestinal parasites | | | |
| Negative | 2762 | 81.07 | (79.72–82.35) |
| Positive | 645 | 18.93 | (17.65–20.28) |

*2.2. Indices of Parasites and Variation in Prevalence of Intestinal Protozoan by Study Year and Age Groups*

Of a total of 3407 fecal samples analyzed, 645 positive samples were found to contain intestinal protozoa in single parasitism, double parasitism, or polyparasitism, representing a single parasite index (SPI) or prevalence of 18.93% (95% CI: 17.65–20.28). Among these confirmed intestinal protozoan infections, 779 strains belonging to nine intestinal protozoan species were enumerated, representing a corrected parasite index (CPI) of 22.96%. The polyparasitism index (PPI) is derived from the difference between CPI and SPI. The PPI in our study is 3.96%. The variation in prevalence over the years showed a significantly growing trend from 3.92% in 2011 to 16.02% in 2012. Between 2012 and 2013, however, it decreased from 16.02% to 10.17%. Following the positive cases from 2013 to 2020, there is a sawtooth pattern (Figure 1).

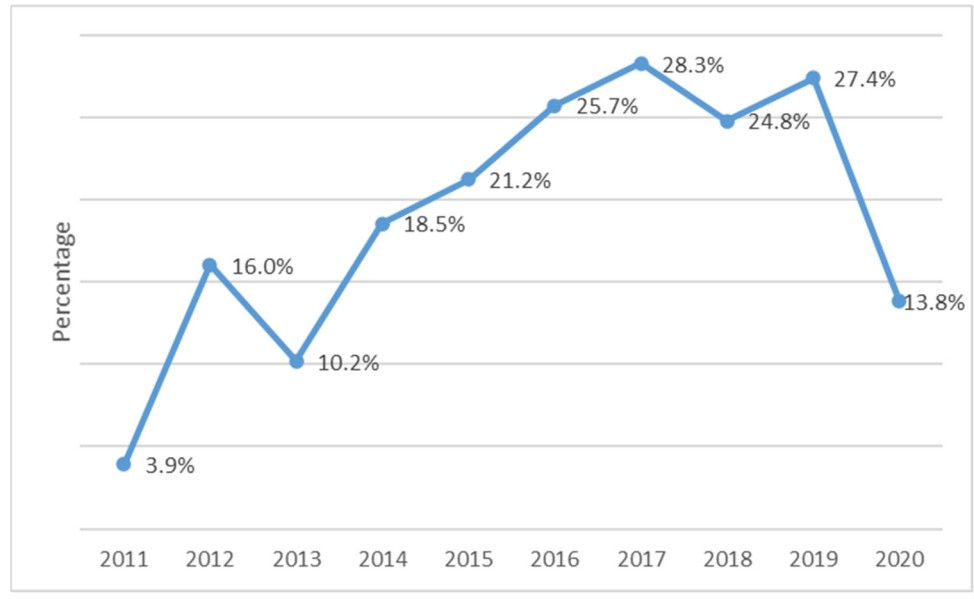

**Figure 1.** Variation in the prevalence of intestinal protozoan infection by year of patients in stool sample analysis from Dakar, Senegal, in the years 2011–2020.

When the distribution of the patients according to age was examined, the highest rate of parasites was 24.83% between the ages of 0–15 years, 19.33% in the individuals aged 31–60 years, 17.365% in the age group 15–30 years, and 16.58% in the individuals aged 60 years and above (Figure 2).

*2.3. Pattern of Identified Species*

Of the total 645 positive samples, 579 (16.99%) were identified in monoparasitism in decreasing order: *Entamoeba coli* (6.87%) and *Blastocystis hominis* (5.69%) for low pathogenic species, and *Entamoeba histolytica/dispar* (2.31%) and *Giardia intestinalis* (1.32%) for pathogenic species (Table 2).

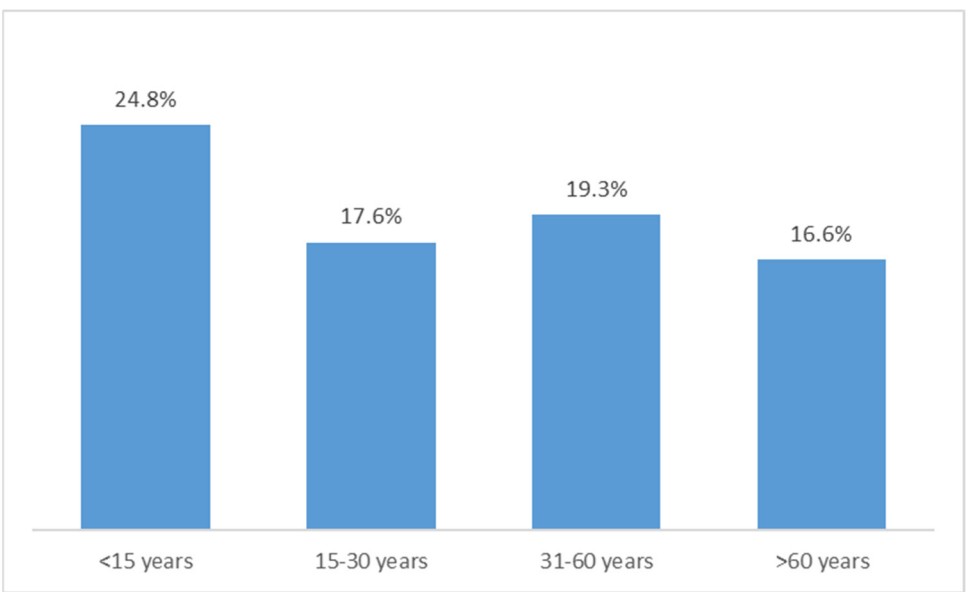

**Figure 2.** Intestinal protozoan infection prevalence by age group of patients in stool samples analysis from Dakar, Senegal, in the years 2011–2020.

**Table 2.** Distribution of species which cause intestinal protozoa in stool sample analysis from Dakar, Senegal, in the years 2011–2020.

| Species | Frequency | Percentage | CI 95% |
|---|---|---|---|
| Negative | 2762 | 81.07 | (79.72–82.35) |
| Monoparasitism | 579 | 16.99 | (15.77–18.29) |
| *Entamoeba coli* | 234 | 6.87 | (06.07–07.77) |
| *Blastocystis hominis* | 194 | 5.69 | (04.96–06.52) |
| *Entamoeba histolytica/dispar* | 79 | 2.31 | (01.86–02.88) |
| *Giardia intestinalis* | 45 | 1.32 | (0.99–01.76) |
| *Trichomonas intestinalis* | 18 | 0.53 | (0.33–0.84) |
| *Endolimax nana* | 4 | 0.12 | (0.04–0.31) |
| *Cystoisospora belli* | 4 | 0.12 | (0.04–0.31) |
| *Cryptosporidium* spp. | 1 | 0.03 | (0.00–0.21) |
| Biparasitism | 64 | 1.88 | (01.47–02.39) |
| *G.intestinalis–E. coli* | 2 | 0.06 | (0.01–0.23) |
| *B. hominis–E. nana* | 1 | 0.03 | (0.00–0.21) |
| *C. belli–B. hominis* | 1 | 0.03 | (0.00–0.21) |
| *E.coli–B.hominis* | 26 | 0.76 | (0.52–01.12) |
| *E.histolytica/dispar–B.hominis* | 7 | 0.21 | (0.10–0.43) |
| *E.histolytica/dispar–E. coli* | 16 | 0.47 | (0.29–0.77) |
| *E.histolytica/dispar–T.intestinalis* | 1 | 0.03 | (0.00–0.21) |
| *G. intestinalis–B. hominis* | 1 | 0.03 | (0.00–0.21) |
| *G. intestinalis–E. coli* | 7 | 0.21 | (0.10–0.43) |
| *T. intestinalis–E. coli* | 1 | 0.03 | (0.00–0.21) |
| *T. intestinalis–B. hominis* | 1 | 0.03 | (0.00–0.21) |
| Polyparasitism | 2 | 0.06 | (0.01–0.23) |
| *E. histolytica/dispar–B. hominis–Chilomastix Mesnili* | 1 | 0.03 | (0.00–0.21) |
| *E.histolytica/dispar–B. hominis–E. coli* | 1 | 0.03 | (0.00–0.21) |

In the case of biparasitism (1.88%), 64 associations were identified. The most common associations were dominated by *Blastocystis hominis–Entamoeba coli* with 26 cases, *Entamoeba coli–Entamoeba histolytica/dispar* with 16 cases, and *Blastocystis hominis–Entamoeba histolytica/dispar* and *Entamoeba coli–Giardia intestinalis* with 9 cases (Table 2).

Only two cases of polyparasitism (0.06%) were reported with Entamoeba histolytica/dispar–Blastocystis hominis–Entamoeba coli and Entamoeba histolytica/dispar–Blastocystis hominis–Chilomastix mesnili (Table 2).

### 2.4. Factors Associated with Intestinal Protozoan Infections

In multivariate analysis from a logistic regression model, protozoan intestinal infections were significantly prevalent in the years 2011 to 2020. Intestinal protozoal infections were significantly more frequent in non-hospitalized patients ($p = 0.0001$). No statistically significant associations were found between the isolated intestinal protozoa and age, gender, and season (Table 3).

**Table 3.** Associated factors with intestinal protozoan infections of patients in stool sample analysis from Dakar, Senegal, in the years 2011–2020.

| | Frequency %) | OR * (95% CI) | *p* Value |
|---|---|---|---|
| Years | | | |
| 2011 | 16 (3.92) | 1 | |
| 2012 | 70 (16.02) | 4.75 (02.70–08.34) | 0.000 |
| 2013 | 35 (10.17) | 2.89 (01.57–05.34) | 0.001 |
| 2014 | 58 (18.53) | 6.51 (03.64–11.65) | 0.000 |
| 2015 | 62 (21.23) | 6.38 (03.59–11.35) | 0.000 |
| 2016 | 107 (25.72) | 8.19 (04.72–14.24) | 0.000 |
| 2017 | 117 (28.26) | 8.88 (05.13–15.37) | 0.000 |
| 2018 | 85 (24.78) | 7.76 (04.42–13.64) | 0.000 |
| 2019 | 69 (27.38) | 8.84 (04.97–15.72) | 0.000 |
| 2020 | 26 (13.83) | 3.75 (01.95–07.21) | 0.000 |
| Age Group | | | |
| <15 Years | 75 (24.83) | 1 | |
| 15–30 Years | 201 (17.65) | 0.98 (00.70–01.33) | 0.841 |
| 31–60 Years | 302 (19.33) | 1.03 (00.76–01.40) | 0.850 |
| >60 Years | 67 (16.58) | 0.84 (00.57–01.23) | 0.378 |
| Gender | | | |
| Male | 311 (17.93) | 1 | |
| Female | 334 (19.98) | 1.11 (00.93–01.33) | 0.233 |
| Service | | | |
| Hospitalized | 113 (12.61) | 1 | |
| Non-Hospitalized | 532 (21.19) | 1.68 (01.33–02.13) | 0.000 |
| Season | | | |
| Dry | 517 (18.37) | 1 | |
| Rainy | 128 (21.59) | 1.16 (00.92–01.47) | 0.207 |

*Adjusted odds ratio. Goodness of fit: Hosmer–Lemeshow $\chi^2$ (8df) = 3.96, $p = 0.8610$.

A statistically significant association existed between age groups and *Giardia intestinalis* and *Blastocystis hominis* ($p < 0.05$). No association was found between species and gender. However, *Entamoeba coli* and *Blastocystis hominis* were more isolated in non-hospitalized patients (Table 4).

Depending on the season, Blastocystis hominis and Trichomonas intestinalis were more isolated in the dry season. Giardia intestinalis, Entamoeba coli, Blastocystis hominis, and Entamoeba histolytica/dispar were identified between 2011 and 2020 (Table 5).

**Table 4.** Prevalence of different intestinal parasites in relation to age groups, gender, and service of patients in stool sample analysis from Dakar, Senegal, in the years 2011–2020.

| Species | Age Group (Years) | | | | | Gender | | | Service | | |
|---|---|---|---|---|---|---|---|---|---|---|---|
| | <15 Years | 13–30 Years | 31–60 Years | >60 Years | *p* Value | Male | Female | *p* Value | Hospitalized | Non-Hospitalized | *p* Value |
| | (N = 302) | (N = 1139) | (N = 1562) | (N = 404) | | (N = 1735) | (N = 1672) | | (N = 896) | (N = 2511) | |
| *Giardia intestinalis* | 18 (6.0%) | 12 (1.1%) | 21 (1.3%) | 4 (1.0%) | 0.000 | 27 (1.6%) | 28 (1.7%) | 0.784 | 14 (1.6%) | 41 (1.6) | 0.886 |
| *Entamoeba coli* | 29 (9.6%) | 87 (7.6%) | 140 (9.0%) | 31 (7.7%) | 0.505 | 139 (8.0%) | 148 (7.9%) | 0.377 | 47 (5.2%) | 240 (9.6%) | 0.000 |
| *Blastocystis hominis* | 32 (10.60%) | 79 (6.94%) | 100 (6.40%) | 22 (5.44%) | 0.038 | 116 (6.69%) | 117 (7.00%) | 0.719 | 28 (3.13%) | 205 (8.16%) | 0.000 |
| *Endolimax nana* | 0 | 1 (0.09%) | 3 (0.19%) | 1 (0.24%) | 0.750 | 3 (0.17%) | 2 (1.20%) | 0.685 | 2 (0.22%) | 3 (0.12%) | 0.486 |
| *Cryptosporidium* spp. | 0 | 0 | 1 (0.06%) | 0 | 0.757 | 0 | 1 (0.06%) | 0.308 | 1 (0.11%) | 0 | 0.094 |
| *Cystoisospora belli* | 0 | 1 (0.09%) | 4 (0.26%) | 0 | 0.461 | 1 (0.06%) | 4 (0.24%) | 0.461 | 2 (0.22%) | 3 (0.12%) | 0.486 |
| *Entamoeba histolytica:dispar* | 8 (2.65%) | 34 (2.99%) | 50 (3.20%) | 13 (3.22%) | 0.955 | 52 (3.00%) | 53 (3017%) | 0.771 | 22 (2.46%) | 83 (3.31%) | 0.206 |
| *Chilomastix mesnili* | 1 (0.33%) | 0 | 0 | 0 | 0.016 | 0 | 1 (0.06%) | 0.308 | 0 | 1 (0.04%) | 0.550 |
| *Trichomonas intestinalis* | 2 (0.66%) | 4 (0.35%) | 12 (0.77%) | 3 (0.74%) | 0.570 | 9 (0.52%) | 12 (0.72%) | 0.458 | 8 (0.89%) | 13 (0.52%) | 0.218 |

**Table 5.** Prevalence of different intestinal parasites in relation to season and year of occurrence of patients in stool sample analysis from Dakar, Senegal, in the years 2011–2020.

| Species | Season | | | Years | | | | | | | | | | *p* Value |
|---|---|---|---|---|---|---|---|---|---|---|---|---|---|---|
| | Dry | Rainy | *p* Value | 2011 | 2012 | 2013 | 2014 | 2015 | 2016 | 2017 | 2018 | 2019 | 2020 | |
| | (N = 2814) | (N = 593) | | (N = 408) | (N = 437) | (N = 344) | (N = 313) | (N = 292) | (N = 416) | (N = 414) | (N = 343) | (N = 252) | (N = 188) | |
| *Giardia intestinalis* | 49 (1.7%) | 6 (1.0%) | 0.200 | 1 (0.3%) | 7 (1.6%) | 2 (0.6%) | 4 (1.3%) | 3 (1.0%) | 13 (3.1%) | 7 (1.7%) | 10 (2.9%) | 6 (2.4%) | 2 (1.1%) | 0.026 |
| *Entamoeba coli* | 239 (8.5%) | 48 (8.1%) | 0.751 | 12 (2.9%) | 41 (9.4%) | 25 (7.3%) | 42 (13.4%) | 29 (9.9%) | 31 (7.5%) | 41 (9.9%) | 34 (9.9%) | 25 (9.9%) | 7 (3.7%) | 0.000 |
| *Blastocystis hominis* | 176 (6.25%) | 57 (9.61%) | 0.003 | 0 | 0 | 0 | 2 (0.64%) | 16 (5.48%) | 60 (14.42%) | 67 (16.18%) | 39 (11.37%) | 35 (13.89%) | 14 (7.45%) | 0.000 |
| *Endolimax nana* | 5 (0.18%) | 0 | 0.304 | 0 | 0 | 0 | 1 (0.32%) | 1 (0.34%) | 0 | 1 (0.24%) | 2 (0.58%) | 0 | 0 | 0.427 |
| *Cryptosporidium* spp. | 0 | 1 (0.17%) | 0.029 | 0 | 0 | 0 | 0 | 0 | 0 | 1 (0.24%) | 0 | 0 | 0 | 0.613 |
| *Cystoisospora belli* | 3 (0.11%) | 2 (0.34%) | 0.182 | 0 | 1 (0.23%) | 0 | 1 (0.32%) | 0 | 0 | 1 (0.24%) | 0 | 0 | 2 (1.06%) | 0.094 |
| *Entamoeba histolytica/dispar* | 85 (3.02%) | 20 (3.37%) | 0.652 | 1 (0.25%) | 22 (5.03%) | 6 (1.74%) | 9 (2.88%) | 13 (4.45%) | 12 (2.88%) | 18 (4.35%) | 6 (1.75%) | 12 (4.76%) | 6 (3.19%) | 0.001 |
| *Chilomastix Mesnili* | 1 (0.03%) | 0 | 0.646 | 0 | 0 | 0 | 0 | 0 | 1 (0.24%) | 0 | 0 | 0 | 0 | 0.617 |
| *Trichomonas intestinalis* | 14 (0.50%) | 7 (1.18%) | 0.053 | 2 (0.49%) | 1 (0.23%) | 3 (0.87%) | 3 (0.96%) | 3 (1.03%) | 3 (0.72%) | 4 (0.97%) | 1 (0.29%) | 1 (0.40%) | 0 | 0.759 |

### 3. Discussion

Protozoa are very important in the etiology of intestinal parasites. Nevertheless, their specific study is poorly developed in Senegal, which justified this study in the parasitology–mycology laboratory of the CHU Le Dantec of Dakar during the following months: January 2011 to December 2020. An overall prevalence of 18.9% was found. This prevalence can be considered as high compared to the one found by same laboratory during the period from January 2011 to December 2015. An overall prevalence of 12.3% was found [14]. Sylla et al., in another laboratory of Dakar, found a prevalence of 22% between 2006 and 2010 at the Fann Hospital. This prevalence of protozoans in these studies conducted in Senegal could be explained by the use of mebendazole in mass administration campaigns. Mebendazole is active on helminths. This may also be related to poor hygiene, which favors the transmission of protozoa, as parasites are more frequent in the wet season than in the dry season [16]. Another study in Senegal in 2020 found a high prevalence of 80.4%. In this study, the authors used molecular methods. They explained that non-molecular methods, such as microscopic observation of fresh feces, are known to significantly underestimate the prevalence of the parasite [17].

In West Africa, an Ivorian study of schoolchildren in the Man region found a prevalence of 98.5% of intestinal protozoal infections [18]. The latter value is much higher than ours. However, the difference can be put into perspective, as this cross-sectional study was conducted in a population aged 6–16 years, where sanitary conditions are much more compromised, especially with promiscuity. Recently, in 2022, Wale and Solomon found a prevalence of 65% in Ethiopia. In this study, the authors investigated risk factors for intestinal protozoal disease. This 65% prevalence could be due to the ingestion of unwashed vegetables, reluctance to wash hands before eating and after using the toilet, accessibility of latrines, and dirty fingernails [19]. A review of 1645 articles reported data from 29,968 school children in Africa, and a pooled prevalence of intestinal protozoan parasites of 25.8% (95% CI: 21.2%–30.3%) was found. In this paper, the prevalences in North Africa, East Africa, Central Africa, West Africa, and South Africa were 40.2%, 21.9%, 21.5%, 32.3%, and 18.6%, respectively [20]. In contrast, in Qatar, the prevalence that was found (5.93%) by a study conducted between 2005 and 2014 was two less than ours. This rather low prevalence compared to ours could be explained by the fact that intestinal parasites are more frequent in undeveloped countries (30–60%) than in advanced countries (2%) [21]. From the results of this study, women (19.98%) had a moderately higher rate of infection than men (17.93%). An inverse with similar proportions was found in Malaysia in a study on intestinal protozoa, with 51% of males versus 49% of females [22]. This probably proves that gender does not really influence the intestinal parasite infestation. In another study [23], women (49.02%) had a somewhat lower rate of infection than men (50.98%), which may be caused by the fact that more men than women are involved in outdoor activities, especially on soil contaminated with feces, such as in agriculture and soccer. A similar finding was reached in other studies that suggested that the propagation of the disease of intestinal protozoan parasites is more common in men than women [24]. Contrary to this, however, Marwa Omar [25] reported that women were more affected by the intestinal protozoan parasite than men. A study conducted among pregnant women revealed that the source of drinking water and occupation (being a farmer) had a statistically significant association with intestinal parasite infection [26].

In this study, protozoan infections were more prevalent in children under 15 years of age, while they decreased with age. The observed prevalence of intestinal protozoan infection may be caused by low individual immunity, lower hand washing sensitization, and other individual hygiene measures in this age group. Akinbo et al. [27] and Hailu and Ayele [28] have noted that age is a possible risk factor for intestinal parasitism. On the contrary, Hussein et al. [29] and Abbaszadeh Afshar et al. [30] disavowed any relationship between age or sex and intestinal parasitism.

In terms of hospitalized or non-hospitalized status, protozoal intestinal infections were significantly more prevalent in non-hospitalized individuals, with 21.19% compared

to 12.61%. This same finding was reached when the epidemiology of intestinal parasitic infections was studied at the Fann Hospital in Dakar [16]. This finding can be interpreted as a result of the fact that patients considered as non-hospitalized (ambulatory) are often hospitalized in other health care facilities without a parasitology laboratory.

The species identified in our studies are still, with few differences, the same as worldwide, but the prevalence of specific species varies over time and from area to area. *E. coli* (6.87%), *B. hominis* (5.69%), *E. histolytica/dispar* (2.31%), and *G. intestinalis* (1.32%) were the most common species found in our series. The mentioned species were also found previously among the prevalent species by El Guamri et al. and Baba et al. in Morocco in 2009 and in Mauritania in 2012, respectively [31,32]. The order of occurrence, however, could be different. *Entamoeba coli* was the most common parasite found, with 6.87%. Nevertheless, it has been reported as an environmental pollution indicator due to poor cleaning and hygiene of the people in the area [33]. *Entamoeba coli* is a commensal parasite that is found in the intestinal tract but does not produce clinically relevant symptoms. It is located only in the intestinal tract lumen, but not in the intestinal epithelial cells. The next most common protozoan isolated in our study was *Blastocystis hominis* (5.69%). A high prevalence of *Blastocystis* infections and subtypes of this species has recently been reported in subjects with close contact with animals and animal handlers [34,35], demonstrating that transmission of the parasite between humans and animals may be common in pastoral farming communities. Among the eight species of intestinal protozoa, the main pathogen was *E. histolytica/dispar* with a prevalence level of 2.31%, followed by *G. intestinalis* (1.32%). Variations in the prevalence rates of *E. histolytica/dispar* and *G. intestinalis* could be attributed to poor sanitation, drinking water source contamination, inadequate hand washing practices, and eating raw vegetables. In our study, *G. intestinalis* was significantly associated with age. The adverse effect of *G. intestinalis* on the development and health of children has been demonstrated by a number of studies [36]. This parasite is known to be responsible for inducing diarrhea and malabsorption syndrome, and can contribute to protein-energy malnutrition, vitamin A deficiency, iron deficiency anemia, and vitamin B12 deficiency anemia [37]. The protozoal associations we identified were characterized by *B. hominis–E. coli*, *E. coli–E histolytica/dispar*, and *E. coli–G. intestinalis* by biparasitism. Two polyparasitisms were found: *E. histolytica/dispar–B.hominis–Chilomaxtix mesnilii* and *E. histolytica/dispar–B.hominis–E. coli*. In Indonesia, Sri Wahdini et al. reported the same combined infections: *B. hominis–E. coli*, *B. hominis–G. intestinalis*, *E. coli–G. intestinalis*, and *E. histolytica/dispar–B. hominis–E. coli* [38].

The associations Identified in our study very often show species regarded as not very or not at all pathogenic, such as *B. hominis*, *E. coli*, or *T. intestinalis*, which indicates the opportunistic and recurrent character of these protozoan species which, in the presence of appropriate factors, can grow in number and cause intestinal disorders.

Our present study has a number of limitations due to the retrospective assessment of intestinal protozoa in the data. Consequently, there are important data that must be analyzed, such as possible risk factors for parasites (hand hygiene, food safety training, medical check-up, educational status, monthly income, hand and vegetable washing, pets and domestic livestock, type of house material, source of drinking water, use of water treatment—chlorine or boiling, use of latrines, and rural/urban residence). The technique of diagnostic stool analysis that is used in Senegal in general is direct wet mount, which may be underestimating the prevalence of enteric protozoa in this retrospective study.

## 4. Methodology

### 4.1. Area and Population Studied

We conducted a descriptive, retrospective study at the CHU Le Dantec parasitology and mycology laboratory in Dakar. Between 2011 and 2020, all patients received at the laboratory were selected for parasitological stool exam showing signs of intestinal protozoal infections. Over the 10-year study period, all patients suspected of having intestinal protozoa in the studied area were selected for inclusion, and all data without demographic

characteristics and the year of fecal examination performed, as well as data without species and stage of the intestinal parasite, were not included.

The principal data collection tool was the bench-top logbooks specifically for parasitological examination of stools. These registers were used to collect data on age, sex, season, hospitalized or not hospitalized status, and year (month) and fecal examination results. The dry season was defined as January to June and October to December. The rainy season was defined as July, August, and September. Age was defined in four categories: children (0–15 years), 16–30 years, 31–60 years, and >60 years.

### 4.2. Fecal Sample Analysis

Fecal samples were sent to the laboratory quickly after collecting them in a plastic bottle for hospitalized patients; they were collected in the laboratory itself for non-hospitalized patients. Fecal samples were analyzed for intestinal parasites using the routine standard procedures used by hospitals and laboratory microbiologists for the identification of parasites [39]. They were initially examined macroscopically by noting their consistency, color, and the existence of blood, mucus, or intestinal adult worms. A physiological saline (0.9%) and Lugol's iodine smear was prepared and examined under a light microscope with 10× and 40× objectives. From each sample, several slides were prepared. For every slide, about 2 mg (the size of a matchstick head) of fecal material was sampled from both the surface and interior of the specimen to improve the rates of detection of the various parasites. Special wet mount slides were used to identify the eggs of helminths and the cysts of protozoa. The samples that did not indicate intestinal parasites by direct smear were examined with the use of the Ritchie concentration technique (formalin–ether concentration). A formalin–ether concentration technique was performed. Approximately 1 to 1.5 g of fecal sample was pooled in a centrifuge tube with 10 mL of formalin mixture and agitated until a suspension was obtained. Next, 3 ml of ether was adding to the suspension and carefully mixed by placing a rubber stopper in the tube and shaking for ten seconds. The tube was inserted into a centrifuge for 2 to 3 minutes at 2000 rpm. Then, the tube was withdrawn from the centrifuge, where 4 layers were seen from top to bottom (top layer of ether, 2nd layer of fat debris, 3rd layer of formalin, and bottom layer of sediment). The first three layers were dumped. A small portion of liquid residue was returned to the sediment, mixed adequately with the deposit, and a drop of sediment was placed on a cleaned slide and coverslip. At last, the slide was examined at 10× and 40× objectives for the search for the intestinal parasites.

### 4.3. Analysis of the Data

We entered the data into Excel and used STATA 10 software to analyze the data. For the descriptive data, percentage was used to evaluate the prevalence of each outcome. Proportions were then compared using the chi-square test or Fisher's exact test (univariate analysis); the significance levels of the tests were 0.05 and two-sided. Stepwise logistic regression was used to determine the association of this pattern with variables, including age, sex, season, and hospitalization/non-hospitalization status. The goodness of fit of the final models was tested using the Hosmer–Lemeshow test of fit.

The formulas below were used to calculate the parasitic indices:

-(i) Simple parasitic index (SPI) is the percentage of subjects parasitized in relation to the total number of fecal parasitological examinations carried out multiplied by one hundred.

-(ii) Corrected parasitic index (CPI) is the ratio of number of parasites identified to the number of total examinations multiplied by one hundred.

-(iii) Polyparasitism index (PPI) is the coexistence in the same individual of two or more parasitic species. The PPI I derived from the difference between the CPI and the SPI.

## 5. Conclusions

The prevention and control of gastrointestinal protozoal infections is now more possible than ever, thanks to the availability of both safe and successful therapeutic drugs and the development and standardization of certain laboratory diagnostic techniques. In the last few years, overall health care policies have focused on community cooperation and preventive medicine in the management of chronic diseases and have created a favorable context for the design and practical implementation of intestinal parasitic infection control measures. The assessment of the results of our hospital laboratory, which is one of the centers that can diagnose many patients, will be a contribution to the epidemiology data of our country. In the perspective of the results from the different regions of our country, our results will be able to properly orient the required strategies for the diagnosis and treatment of intestinal parasitic infections and the establishment of preventive actions. This underlines the fact that parasite infections still represent an essential public health issue.

**Author Contributions:** Conception, M.N.; Analysis, M.N.; Methods, M.N.; Drafting—Original Version, M.N.; Drafting—Revision and Editing, K.D., M.A.D., D.N., A.S.B., M.C.S. and E.K. All authors have read and agreed to the published version of the manuscript.

**Funding:** This research received no external funding.

**Institutional Review Board Statement:** Names of individuals were removed from the data acquired from the laboratory to protect privacy, and only unique ID numbers were used in order to identify individuals. We attest that the study was performed in conformity with ethical guidelines as defined in the 1964 Declaration of Helsinki and its subsequent amendments.

**Informed Consent Statement:** This was a retrospective study.

**Data Availability Statement:** Not applicable.

**Acknowledgments:** A special thank you is given to all the laboratory technical staff for their helpful and devoted assistance in the study implementation and their dexterous work conducted in the laboratory.

**Conflicts of Interest:** The author(s) indicate no conflict of interest. The authors are exclusively responsible for the design and drafting of this article.

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
