# Peer review of "Retrospective Assessment of The Intestinal Protozoan Distribution in Patients Admitted to The Hospital Aristide Le Dantec in Dakar, Senegal, from 2011 to 2020"

_parasitologia, doi:10.3390/parasitologia3010001_

Round 1
Reviewer 1 Report
The article entitled "Retrospective evaluation of the disctribution of intestinal protozoan in patients admitted to Le Dantec University Hospital of Dakar, Senegal between 2011-2020" is an interesting clinical survey-based study. However, this article should be revise enough to get an opportunity to be published in Parasitologia.
Authors have taken 3407 no of people which is highly commendable however, the sex ratio they have mentioned is not clear is it male to female or the vice versa? However, in due course the number has been mentioned in the table and from there we can calculate which is not at all appreciated. Kindly mention that in the text as well.
Authors have meticulously performed the statistical surveys and what I feel this paper should be written in somewhat more interesting way. The approach should be more clear. There is some sort of monotony in the entire text. Authors are requested to present their data differently.
Tables are comprehensive, however, some pictorial representations should be given in order to attract readers.
Author Response
Response to Reviewer 1 Comments
Dear Reviewers,
We thank all the reviewers of our manuscript whose comments and suggestions will certainly contribute to the improvement of its quality.
All the corrections raised by the different reviewers were directly made in the text (highlight in yellow) and the answers to the different questions and suggestions are mentioned in red below.
Point 1: Must be improved: Does the introduction provide sufficient background and include all relevant references?
Response 1: we have improved the introduction by mentioning the WHO guidelines for the treatment of intestinal parasitosis, in particular protozoosis.
“To achieve the goal of eliminating intestinal parasitic infections as a public health problem, the WHO has suggested the mass administration of a single oral dose of mebendazole or albendazole administered periodically to preschool and school-aged children living in endemic areas. This is an intervention that reduces morbidity by decreasing the vermin burden [ 10,11].
There are few or no effective drug treatments for intestinal protozoa. Treatment of Giardia and amoebae is based on 5-nitroimidazole derivatives. Single-dose treatments can be administered with tinidazole or secnidazole [12].”
10- Levecke B, Montresor A, Albonico M, Ame SM, Behnke JM, Bethony JM, et al. Assessment of anthelmintic efficacy of mebendazole in school children in six countries where soil-transmitted helminths are endemic. PLoS Negl Trop Dis. 2014;8(10): e3204.
11- Gabrielli AF, Montresor A, Chitsulo L, Engels D, Savioli L. Preventive chemotherapy in human helminthiasis: theoretical and operational aspects. Trans R Soc Trop Med Hyg. 2011;105(12):683-693.
12- Loiseau PM, Le Bras J. [New drugs against parasitic diseases]. Rev Prat. 2007 Jan 31;57(2):175-82.
Point 2: The sex ratio they have mentioned is not clear is it male to female or the vice versa?
Response 2: the mentioned sex ratio is calculated by ratio Male/Female. it is 1735/1672=1. 037.Rounded to 1.04.
Point 3: Authors are requested to present their data differently.
Response 3:
Table 1: Socio-demographic characteristics of participants
|
Number |
Percentage |
CI 95% |
Years |
|
|
|
2011 |
408 |
11.98 |
(10.93-13.11) |
2012 |
437 |
12.83 |
(11.74-13.99) |
2013 |
344 |
10.1 |
(09.13-11.15) |
2014 |
313 |
9.19 |
(08.26-10.20) |
2015 |
292 |
8.57 |
(07.68-09.56) |
2016 |
416 |
12.21 |
(11.15-13.35) |
2017 |
414 |
12.15 |
(11.10-13.29) |
2018 |
343 |
10.07 |
(09.10-11.12) |
2019 |
252 |
7.4 |
(06.56-08.33) |
2020 |
188 |
5.52 |
(04.80-06.34) |
Age group |
|
||
<15 yrs |
302 |
8.86 |
(07.95-09.87) |
15-30 yrs |
1,139 |
33.43 |
(31.87-35.03) |
31-60 yrs |
1,562 |
45.85 |
(44.18-47.52) |
>60 yrs |
404 |
11.86 |
(10.81-12.99) |
Gender |
|
||
Male |
1,735 |
50.92 |
(49.24-52.60) |
Female |
1,672 |
49.08 |
(47.40-50.76) |
Service |
|
||
Hospitalized |
896 |
26.3 |
(24.85-27.8) |
Nonhospitalized |
2,511 |
73.7 |
(72.2-75.15) |
Seasons |
|
||
Dry |
2,814 |
82.59 |
(81.28-83.83) |
Rainy |
593 |
17.41 |
(16.17-18.72) |
Intestinal parasites |
|
||
Negative |
2,762 |
81.07 |
(79.72-82.35) |
Positive |
645 |
18.93 |
(17.65-20.28) |
It was observed that 50.92% of the patients were male and 49.08% were female. When the distribution of the patients according to season was examined, the dry season presented highest rate of 82.59% compared with rainy season (17.41%). According the service, nonhospitalized (73.70%) were more represented than hospitalized (26.30%).
Figure 1: Evolution of the intestinal parasitic infection’s prevalence by year of occurrence
We added in the results a figure 2 (Figure 2: Intestinal protozoan infection’s prevalence by age group)
When the distribution of the patients according to age was examined, the highest rate of parasites was 24.83% between the ages of 0-15 years, 19.33% in the individuals aged 31-60 years, 17.365% in age group 15-30 years and 16.58% in the individuals aged 60 years and above (Figure 2).
Figure 2: Intestinal protozoan infection’s prevalence by age group
Table 2: Species distribution
Species |
Frequency |
Percentage |
CI 95% |
NEGATIVE |
2762 |
81.07 |
(79.72-82.35) |
MONOPARASITISM |
579 |
16.99 |
(15.77-18.29) |
Entamoeba coli |
234 |
6.87 |
(06.07-07.77) |
Blastocystis hominis |
194 |
5.69 |
(04.96-06.52) |
Entamoeba histolytica/dispar |
79 |
2.31 |
(01.86-02.88) |
Giardia intestinalis |
45 |
1.32 |
(0.99-01.76) |
Trichomonas intestinalis |
18 |
0.53 |
(0.33-0.84) |
Endolimax nana |
4 |
0.12 |
(0.04-0.31) |
Cystoisospora belli |
4 |
0.12 |
(0.04-0.31) |
Cryptosporidium spp |
1 |
0.03 |
(0.00-0.21) |
BIPARASITISM |
64 |
1.88 |
(01.47-02.39) |
G.intestinalis -E.coli |
2 |
0.06 |
(0.01-0.23) |
B. hominis-E. nana |
1 |
0.03 |
(0.00-0.21) |
C. belli-B. hominis |
1 |
0.03 |
(0.00-0.21) |
E.coli -B. hominis |
26 |
0.76 |
(0.52-01.12) |
E. histolytica/dispar-B. hominis |
7 |
0.21 |
(0.10-0.43) |
E. histolytica/dispar - E. coli |
16 |
0.47 |
(0.29-0.77) |
E. histolytica/dispar-T. intestinalis |
1 |
0.03 |
(0.00-0.21) |
G. intestinalis- B. hominis |
1 |
0.03 |
(0.00-0.21) |
G. intestinalis- E. coli |
7 |
0.21 |
(0.10-0.43) |
T. intestinalis - E. coli |
1 |
0.03 |
(0.00-0.21) |
T. intestinalis- B. hominis |
1 |
0.03 |
(0.00-0.21) |
POLYPARASITISM |
2 |
0.06 |
(0.01-0.23) |
E. histolytica/dispar-B. hominis -Chilomastix Mesnili |
1 |
0.03 |
(0.00-0.21) |
E. histolytica/dispar-B. hominis -E.coli |
1 |
0.03 |
(0.00-0.21) |
Point 4: (x) English language and style are fine/minor spell check required
Response 4: We propose that our manuscript be checked by a native English speaker after correcting the reviewers' comments and suggestions
Reviewer 2 Report
- Lines 12-29: Alignment of abstract and keywords is outside the journal's standard.
- The abstract has 287 words, 87 more than what is allowed in the guidelines for authors.
- Line 17: Entamoeba hitolytica must be in italic.
- Line 18: The phrase "128 parasites (64 associations) were identified" is not understandable. It is not clear from the text what kind of associations the authors refer to.
- Lines 24-25: The authors state an association with the age of the patients, without, however, mentioning which age group is most susceptible to the occurrence.
- Keywords are of low quality as they are terms primarily found in the title of the manuscript. Suggestions: Entamoeba, Giardia, Hygiene, Protozoan Infections, Stool Samples.
- All topic and subtopic titles are misaligned.
- Line 39: Cryptosporidium must be in italic. Replace "sp" with "spp."
- Line 40: Genus of "G. intestinalis", "B. hominis", and "C. cayatanensis" must not be abbreviated, because is the first mentions of these species in the main text.
- Line 41-42: The authors mentioned food contamination as a risk for protozoan infection occurrence, but forget the importance of waterborne infection, as occurs for Cryptosporidium spp, cited in line 39. Increase the information on the background.
- Line 46-47: Remove "our hospital", and replace with the name of the institution and its location.
- Table 1 caption should be improve. Remember that any Table must be understood by a reader without a main text. Example: Socio-demographic profiles of patients in an stool sample analysis from Dakar, Senegal, in the years 2011-2020. The first column of Table 1 can also be left-aligned.
- Figure 1 caption, also should be improve. Remember that any Table must be understood by a reader without a main text.
- Line 71: Replace "645 protozoa" with "645 positive samples"
- In sub-item 2.2, why were confidence intervals not calculated for each infection? Table 2a and 2b should have a column for the presentation of confidence intervals.
- Line 72: Blastocystis hominis must be in italic.
- Line 73: Replace "(Table 2(a))" with "(Table 2a)"
- In the main text of sub-item 2.2, all genus in the species names mention before must be abbreviated.
- Line 75: Entamoeba coli-Entamoeba histolytica/dispar must be in italic.
- Line 78: Elsewhere in the manuscript, the term used is polyparasitism rather than tripparasitism. It is necessary to standardize.
- The column "Total of species" in Tables 2b and 2c is meaningless. I strongly recommend its removal.
- Table 2 (a,b,c) caption should be improve. Remember that any Table must be understood by a reader without a main text.
- Table 3 caption should be improve. Remember that any Table must be understood by a reader without a main text.
- The results of the risk analysis are unclear. There is no accurate interpretation of the results. How are all age groups risk associated? If all age groups are associated risks, then not a greater risk for any of them. It makes no sense. Honestly, the data in Table 3 are quite confusing, and there is no interpretation of these data by the authors.
- The percentage values in Table 3 seem to be very wrong.
- Table 4 caption should be improve. Remember that any Table must be understood by a reader without a main text.
- Table 5 caption should be improve. Remember that any Table must be understood by a reader without a main text.
- All protozoan species names in the main text of the results need to be verified if they have been mentioned previously, to have the genus name abbreviated if they have already been mentioned in the main text. The same in the discussion, where several scientific names of the species are not in italics either.
- Line 150-156: The authors compare the prevalence of the study with other studies, however, they do not mention reasons for the divergent prevalences in the last part of the paragraph. More than presenting data from other studies, it is necessary to say what differences between them (method, population, period, etc.) lead to the differences or similarities found.
- Line 160-161: The sentence "Recently in 2022, Wale and Solomon found a prevalence of 65% in Ethiopia [18]." is loose, like information thrown into the text.
- Line 174-175: Is there any reason in the Marwa-Omar study for women to be more parasitized than men? This needs to be discussed in the text if there are sociocultural reasons mentioned by the authors for this study to have found such significance.
- Line 176-244: This paragraph is impossible to read. Understanding the results is confusing. The size of the paragraph is surreal. This part needs to be rewritten in its entirety.
- References do not follow the journal's standard.
- Formatting needs to be revised.
Author Response
Response to Reviewer 2 Comments
Dear Reviewers,
We thank all the reviewers of our manuscript whose comments and suggestions will certainly contribute to the improvement of its quality.
All the corrections raised by the different reviewers were directly made in the text (highlight in yellow) and the answers to the different questions and suggestions are mentioned in red below.
Point 1: (x) English language and style are fine/minor spell check required
Response 1: We propose that our manuscript be checked by a native English speaker after correcting the reviewers' comments and suggestions
Point 2: Are the results clearly presented? (Must be improved)
Response 2: We have re-analyzed our data
Point 3: Does the introduction provide sufficient background and include all relevant references? (Must be improved)
Response 3: we have improved the introduction by mentioning the WHO guidelines for the treatment of intestinal parasitosis, in particular protozoosis.
“To achieve the goal of eliminating intestinal parasitic infections as a public health problem, the WHO has suggested the mass administration of a single oral dose of mebendazole or albendazole administered periodically to preschool and school-aged children living in endemic areas.This is an intervention that reduces morbidity by decreasing the vermin burden[10,11].
There are few or no effective drug treatments for intestinal protozoa. Treatment of Giardia and amoebae is based on 5-nitroimidazole derivatives. Single-dose treatments can be administered with tinidazole or secnidazole [12].”
10- Levecke B, Montresor A, Albonico M, Ame SM, Behnke JM, Bethony JM, et al. Assessment of anthelmintic efficacy of mebendazole in school children in six countries where soil-transmitted helminths are endemic. PLoS Negl Trop Dis. 2014;8(10): e3204.
11- Gabrielli AF, Montresor A, Chitsulo L, Engels D, Savioli L. Preventive chemotherapy in human helminthiasis: theoretical and operational aspects. Trans R Soc Trop Med Hyg. 2011;105(12):683-693.
12- Loiseau PM, Le Bras J. [New drugs against parasitic diseases]. Rev Prat. 2007 Jan 31;57(2):175-82.
Point 4: - Lines 12-29: Alignment of abstract and keywords is outside the journal's standard.
- The abstract has 287 words, 87 more than what is allowed in the guidelines for authors.
Response 4: New abstract and keywords corrected lines 12-31
Abstract:
Infectious parasites, especially the intestinal protozoan parasites, continue to be a major public health problem in Africa, where many of the same factors contribute to the transmission of these parasites. This study was conducted to investigate the parasites causing intestinal protozoal infections diagnosed in Aristide Le Dantec hospital (Senegal). Direct examination and the Ritchie technique were used. Among the 3407 stool samples studied, 645 demonstrated the presence of intestinal protozoa in single-parasitism, parasitism, or polyparasitism, representing a prevalence of 18.93%. Out of a total of 645 protozoa, 579 (16.99%) were identified in monoparasitism in the following order: Entamoeba coli (6.87%), Blastocystis hominis (5.69%) for low pathogenic species, Entamoeba histolytica/dispar (2.31.%) and Giardia intestinalis(1.32%) for pathogenic species. the rates of parasitism and polyparasitism were 1.88% and 0.06% respectively. The highest rate of parasites was 24.83% between the ages of 0-15 years, A logistical regression model indicated that intestinal protozoan infections were no associated with age groups. There was an association between age groups and Giardia intestinalis, Blastocystis hominis (p<0.05). These results demonstrated the frequency of intestinal protozoa in Senegal. The need to implement treatment, prevention and control measures would limit the circulation of these protozoan infections.
Keywords: Entamoeba histolytica; Giardia intestinalis, Protozoan intestinal; hygiene; Stool samples
Point 5: - Line 39: Cryptosporidium must be in italic. Replace "sp" with "spp."
Response 5: Corrected in line 44
Point 6- Line 40: Genus of "G. intestinalis", "B. hominis", and "C. cayatanensis" must not be abbreviated, because is the first mentions of these species in the main text.
Response 6: Corrected in line 45
Point 7: Line 41-42: The authors mentioned food contamination as a risk for protozoan infection occurrence, but forget the importance of waterborne infection, as occurs for Cryptosporidium spp, cited in line 39. Increase the information on the background.
Response 7: Corrected in line 48
Point 8: - Line 46-47: Remove "our hospital", and replace with the name of the institution and its location.
Response 8: Corrected in line 62
Point 9: - Table 1 caption should be improved. Remember that any Table must be understood by a reader without a main text. Example: Socio-demographic profiles of patients in a stool sample analysis from Dakar, Senegal, in the years 2011-2020. The first column of Table 1 can also be left-aligned.
Response 9: Corrected in line 73-75,
Table 1: Socio-demographic profiles of patients in a stool sample analysis from Dakar, Senegal, in the years 2011-2020
|
Number |
Percentage |
CI 95% |
Years |
|
|
|
2011 |
408 |
11.98 |
(10.93-13.11) |
2012 |
437 |
12.83 |
(11.74-13.99) |
2013 |
344 |
10.1 |
(09.13-11.15) |
2014 |
313 |
9.19 |
(08.26-10.20) |
2015 |
292 |
8.57 |
(07.68-09.56) |
2016 |
416 |
12.21 |
(11.15-13.35) |
2017 |
414 |
12.15 |
(11.10-13.29) |
2018 |
343 |
10.07 |
(09.10-11.12) |
2019 |
252 |
7.4 |
(06.56-08.33) |
2020 |
188 |
5.52 |
(04.80-06.34) |
Age group |
|
||
<15 yrs |
302 |
8.86 |
(07.95-09.87) |
15-30 yrs |
1,139 |
33.43 |
(31.87-35.03) |
31-60 yrs |
1,562 |
45.85 |
(44.18-47.52) |
>60 yrs |
404 |
11.86 |
(10.81-12.99) |
Gender |
|
||
Male |
1,735 |
50.92 |
(49.24-52.60) |
Female |
1,672 |
49.08 |
(47.40-50.76) |
Service |
|
||
Hospitalized |
896 |
26.3 |
(24.85-27.8) |
Nonhospitalized |
2,511 |
73.7 |
(72.2-75.15) |
Seasons |
|
||
Dry |
2,814 |
82.59 |
(81.28-83.83) |
Rainy |
593 |
17.41 |
(16.17-18.72) |
Intestinal parasites |
|
||
Negative |
2,762 |
81.07 |
(79.72-82.35) |
Positive |
645 |
18.93 |
(17.65-20.28) |
Point 10: - Figure 1 caption, also should be improve. Remember that any Table must be understood by a reader without a main text.
Response 7: We have taken figure 1 and added a figure 2 that shows the prevalence of intestinal protozoosis according to age groups in the results
Figure 1: Variation in the prevalence of intestinal protozoan infection by year
When the distribution of the patients according to age was examined, the highest rate of parasites was 24.83% between the ages of 0-15 years, 19.33% in the individuals aged 31-60 years, 17.365% in age group 15-30 years and 16.58% in the individuals aged 60 years and above (Figure 2). Line 99-102
Figure 2: Intestinal protozoan infection’s prevalence by age group
Point 11: - Line 71: Replace "645 protozoa" with "645 positive samples
Response 11: Corrected in line 87 and line 113
Point 12: - In sub-item 2.2, why were confidence intervals not calculated for each infection? Table 2a and 2b should have a column for the presentation of confidence intervals
Response 12: we have resumed the table of the distribution of the species by calculating the confidence intervals
Table 2: Species distribution
Species |
Frequency |
Percentage |
CI 95% |
NEGATIVE |
2762 |
81.07 |
(79.72-82.35) |
MONOPARASITISM |
579 |
16.99 |
(15.77-18.29) |
Entamoeba coli |
234 |
6.87 |
(06.07-07.77) |
Blastocystis hominis |
194 |
5.69 |
(04.96-06.52) |
Entamoeba histolytica/dispar |
79 |
2.31 |
(01.86-02.88) |
Giardia intestinalis |
45 |
1.32 |
(0.99-01.76) |
Trichomonas intestinalis |
18 |
0.53 |
(0.33-0.84) |
Endolimax nana |
4 |
0.12 |
(0.04-0.31) |
Cystoisospora belli |
4 |
0.12 |
(0.04-0.31) |
Cryptosporidium spp |
1 |
0.03 |
(0.00-0.21) |
BIPARASITISM |
64 |
1.88 |
(01.47-02.39) |
G.intestinalis -E.coli |
2 |
0.06 |
(0.01-0.23) |
B. hominis-E. nana |
1 |
0.03 |
(0.00-0.21) |
C. belli-B. hominis |
1 |
0.03 |
(0.00-0.21) |
E.coli -B. hominis |
26 |
0.76 |
(0.52-01.12) |
E. histolytica/dispar-B. hominis |
7 |
0.21 |
(0.10-0.43) |
E. histolytica/dispar - E. coli |
16 |
0.47 |
(0.29-0.77) |
E. histolytica/dispar-T. intestinalis |
1 |
0.03 |
(0.00-0.21) |
G. intestinalis- B. hominis |
1 |
0.03 |
(0.00-0.21) |
G. intestinalis- E. coli |
7 |
0.21 |
(0.10-0.43) |
T. intestinalis - E. coli |
1 |
0.03 |
(0.00-0.21) |
T. intestinalis- B. hominis |
1 |
0.03 |
(0.00-0.21) |
POLYPARASITISM |
2 |
0.06 |
(0.01-0.23) |
E. histolytica/dispar-B. hominis -Chilomastix Mesnili |
1 |
0.03 |
(0.00-0.21) |
E. histolytica/dispar-B. hominis -E.coli |
1 |
0.03 |
(0.00-0.21) |
Point 13: - Line 72: Blastocystis hominis must be in italic
Response 13: Corrected in line 114
Point 14- In the main text of sub-item 2.2, all genus in the species names mention before must be abbreviated.
Response 14: Corrected in line 125 Table 2
Point 15: - Line 75: Entamoeba coli-Entamoeba histolytica/dispar must be in italic.
Response 15: Corrected in line 119
Point 16: - Line 78: Elsewhere in the manuscript, the term used is polyparasitism rather than tripparasitism. It is necessary to standardize.
Response 16: Corrected in line 121
Point 17: - The results of the risk analysis are unclear. There is no accurate interpretation of the results. How are all age groups risk associated? If all age groups are associated risks, then not a greater risk for any of them. It makes no sense. Honestly, the data in Table 3 are quite confusing, and there is no interpretation of these data by the authors.
Response 17: The risk analysis results were recalculated (Table 3)
In multivariate analysis from a logistic regression model, protozoan intestinal infections were significantly prevalent in the years 2011 to 2020. Intestinal protozoal infections were significantly more frequent in nonhospitalized patients (p= 0.0001). No statistically significant associations were found between intestinal protozoa isolated like age, gender and season (Table 3).
Table 3. Associated factors with intestinal protozoan infections.
|
Frequency %) |
OR (95% CI) |
P-value |
Years |
|
|
|
2011 |
16 (3.92) |
1 |
|
2012 |
70 (16.02) |
4.75 (02.70-08.34) |
0.000 |
2013 |
35 (10.17) |
2.89 (01.57-05.34) |
0.001 |
2014 |
58 (18.53) |
6.51 (03.64-11.65) |
0.000 |
2015 |
62 (21.23) |
6.38 (03.59-11.35) |
0.000 |
2016 |
107 (25.72) |
8.19 (04.72-14.24) |
0.000 |
2017 |
117 (28.26) |
8.88 (05.13-15.37) |
0.000 |
2018 |
85 (24.78) |
7.76 (04.42-13.64) |
0.000 |
2019 |
69 (27.38) |
8.84 (04.97-15.72) |
0.000 |
2020 |
26 (13.83) |
3.75 (01.95-07.21) |
0.000 |
Age group |
|
|
|
<15 years |
75 (24.83) |
1 |
|
15-30 years |
201 (17.65) |
0.98 (00.70-01.33) |
0.841 |
31-60 years |
302 (19.33) |
1.03 (00.76-01.40) |
0.850 |
>60 years |
67 (16.58) |
0.84 (00.57-01.23) |
0.378 |
Gender |
|
|
|
Male |
311 (17.93) |
1 |
|
Female |
334 (19.98) |
1.11 (00.93-01.33) |
0.233 |
Service |
|
|
|
Hospitalized |
113 (12.61) |
1 |
|
Nonhospitalized |
532 (21.19) |
1.68 (01.33-02.13) |
0.000 |
Season |
|
|
|
Dry |
517 (18.37) |
1 |
|
Rainy |
128 (21.59) |
1.16 (00.92-01.47) |
0.207 |
*Adjusted odds ratio. Goodness of fit: Hosmer-Lemeshow (8df) = 3.96, P = 0.8610
Point 18: - Line 150-156: The authors compare the prevalence of the study with other studies; however, they do not mention reasons for the divergent prevalences in the last part of the paragraph. More than presenting data from other studies, it is necessary to say what differences between them (method, population, period, etc.) lead to the differences or similarities found.
Response 18: Corrected in line 190-198
Point 19: - Line 160-161: The sentence "Recently in 2022, Wale and Solomon found a prevalence of 65% in Ethiopia [18]." is loose, like information thrown into the text.
Response 19: In this study, the authors investigated risk factors for intestinal protozoal disease. This 65% prevalence could be the ingestion of unwashed vegetables, reluctance to wash hands before eating and after using the toilet, accessibility of latrines, and dirty fingernails. Line 204-207.
Point 20: - Line 174-175: Is there any reason in the Marwa-Omar study for women to be more parasitized than men? This needs to be discussed in the text if there are sociocultural reasons mentioned by the authors for this study to have found such significance.
Response 20: The authors did not explain the high frequency of women. But a study conducted among pregnant women (Ethiopia 2022) revealed that the source of drinking water and occupation (being a farmer) had a statistically significant association with intestinal parasite infection. Corrected in line 227-229
Point 21- Line 176-244: This paragraph is impossible to read. Understanding the results is confusing. The size of the paragraph is surreal. This part needs to be rewritten in its entirety.
Response 20: This section has been rewritten in its entirety. Line 230-279
Point 22- References do not follow the journal's standard.
Response 22: References were corrected
Point 23: Formatting needs to be revised.
Response 23: Formatting revised
Round 2
Reviewer 1 Report
Suggested modifications have been incorporated in the present manuscript. I am happy to accept this for publication.
Author Response
Response to Reviewer 2 Comments
Dear Reviewers,
We thank all the reviewers of our manuscript whose comments and suggestions will certainly contribute to the improvement of its quality.
All the corrections raised by the different reviewers were directly made in the text (highlight in yellow) and the answers to the different questions and suggestions are mentioned in red below.
Point 1: (x) English language and style are fine/minor spell check required
Response 1: We propose that our manuscript be checked by a native English speaker after correcting the reviewers' comments and suggestions
Reviewer 2 Report
First, I would like to thank the authors for the opportunity to collaborate with the manuscript. I have just two considerations: In the legends of figures and tables, insert in each one, the place and period of the data presented, similarly to what was done in the legend of Table 1. The formatting of the legend of Table 4 is in italics and outside of format, needing to be corrected.
Author Response
Response to Reviewer 2 Comments
Dear Reviewers,
We thank all the reviewers of our manuscript whose comments and suggestions will certainly contribute to the improvement of its quality.
All the corrections raised by the different reviewers were directly made in the text (highlight in yellow) and the answers to the different questions and suggestions are mentioned in red below.
Point 1: (x) I don't feel qualified to judge about the English language and style
Response 1: We propose that our manuscript be checked by a native English speaker after correcting the reviewers' comments and suggestions
Point 2: First, I would like to thank the authors for the opportunity to collaborate with the manuscript. I have just two considerations: In the legends of figures and tables, insert in each one, the place and period of the data presented, similarly to what was done in the legend of Table 1. The formatting of the legend of Table 4 is in italics and outside of format, needing to be corrected.
Response 2
Figure 1: Variation in the prevalence of intestinal protozoan infection by year of patients in stool samples analysis from Dakar, Senegal, in the years 2011-2020. Line 323-324
Figure 2: Intestinal protozoan infection’s prevalence by age group of patients in stool samples analysis from Dakar, Senegal, in the years 2011-2020. Line 334-335
Table 2: Distribution of species which cause intestinal protozoa of patients in stool samples analysis from Dakar, Senegal, in the years 2011-2020. Line 348-349
Table 3. Associated factors with intestinal protozoan infections of patients in stool samples analysis from Dakar, Senegal, in the years 2011-2020. Line 469-470
Table 4: Prevalence of different intestinal parasites in relation to age groups, gender and service of patients in stool samples analysis from Dakar, Senegal, in the years 2011-2020. Line 683-684
Table 5. Prevalence of different intestinal parasites in relation to season and year of occurrence of patients in stool samples analysis from Dakar, Senegal, in the years 2011-2020 Line 717
